# Actor-Critic Policy Optimization in Partially Observable Multiagent Environments

**Sriram Srinivasan**[*,1]  **Marc Lanctot**[*,1]  **Vinicius Zambaldi**[1]  **Julien Pérolat**[1]
srsrinivasan@           lanctot@              vzambaldi@               perolat@

**Karl Tuyls**[1]              **Rémi Munos**[1]              **Michael Bowling**[1]
karltuyls@                   munos@                        bowlingm@

...@google.com. [1]DeepMind. [*]These authors contributed equally.

## Abstract

Optimization of parameterized policies for reinforcement learning (RL) is an important and challenging problem in artificial intelligence. Among the most common approaches are algorithms based on gradient ascent of a score function representing discounted return. In this paper, we examine the role of these policy gradient and actor-critic algorithms in partially-observable multiagent environments. We show several candidate policy update rules and relate them to a foundation of regret minimization and multiagent learning techniques for the one-shot and tabular cases, leading to previously unknown convergence guarantees. We apply our method to *model-free* multiagent reinforcement learning in adversarial sequential decision problems (zero-sum imperfect information games), using RL-style function approximation. We evaluate on commonly used benchmark Poker domains, showing performance against fixed policies and empirical convergence to approximate Nash equilibria in self-play with rates similar to or better than a baseline model-free algorithm for zero-sum games, without any domain-specific state space reductions.

## 1   Introduction

There has been much success in learning parameterized policies for sequential decision-making problems. One paradigm driving progress is deep reinforcement learning (Deep RL), which uses deep learning [52] to train function approximators that represent policies, reward estimates, or both, to learn directly from experience and rewards [85]. These techniques have learned to play Atari games beyond human-level [60], Go, chess, and shogi from scratch [82, 81], complex behaviors in 3D environments [59, 97, 37], robotics [27, 73], character animation [67], among others.

When multiple agents learn simultaneously, policy optimization becomes more complex. First, each agent's environment is non-stationary and naive approaches can be non-Markovian [58], violating the requirements of many traditional RL algorithms. Second, the optimization problem is not as clearly defined as maximizing one's own expected reward, because each agent's policy affects the others' optimization problems. Consequently, game-theoretic formalisms are often used as the basis for representing interactions and decision-making in multiagent systems [17, 79, 64].

Computer poker is a common multiagent benchmark domain. The presence of partial observability poses a challenge for traditional RL techniques that exploit the Markov property. Nonetheless, there has been steady progress in poker AI. Near-optimal solutions for heads-up limit Texas Hold'em were found with tabular methods using state aggregation, powered by policy iteration algorithms based on regret minimization [102, 87, 12]. These approaches were founded on a basis of counterfactual

regret minimization (CFR), which is the root of recent advances in no-limit, such as Libratus [16] and DeepStack [61]. However, (i) both required Poker-specific domain knowledge, and (ii) neither were model-free, and hence are unable to learn directly from experience, without look-ahead search using a perfect model of the environment.

In this paper, we study the problem of multiagent reinforcement learning in adversarial games with partial observability, with a focus on the model-free case where agents (a) do not have a perfect description of their environment (and hence cannot do a priori planning), (b) learn purely from their own experience without explicitly modeling the environment or other players. We show that actor-critics reduce to a form of regret minimization and propose several policy update rules inspired by this connection. We then analyze the convergence properties and present experimental results.

## 2  Background and Related Work

We briefly describe the necessary background. While we draw on game-theoretic formalisms, we align our terminology with RL. We include clarifications in Appendix A[1]. For details, see [79, 85].

### 2.1  Reinforcement Learning and Policy Gradient Algorithms

An agent acts by taking actions $a \in \mathcal{A}$ in states $s \in \mathcal{S}$ from their policy $\pi : s \rightarrow \Delta(\mathcal{A})$, where $\Delta(X)$ is the set of probability distributions over $X$, which results in changing the state of the environment $s_{t+1} \sim \mathcal{T}(s_t, a_t)$; the agent then receives an observation $o(s_t, a_t, s_{t+1}) \in \Omega$ and reward $R_t$.[2] A sum of rewards is a **return** $G_t = \sum_{t'=t}^{\infty} R_{t'}$, and aim to find $\pi^*$ that maximizes expected return $\mathbb{E}_\pi[G_0]$.[3]

Value-based solution methods achieve this by computing estimates of $v_\pi(s) = \mathbb{E}_\pi[G_t \mid S_t = s]$, or $q_\pi(s, a) = \mathbb{E}_\pi[G_t \mid S_t = s, A_t = a]$, using temporal difference learning to bootstrap from other estimates, and produce a series of $\epsilon$-greedy policies $\pi(s, a) = \epsilon/|\mathcal{A}| + (1 - \epsilon)\mathbb{I}(a = \mathrm{argmax}_{a'} q_\pi(s, a'))$. In contrast, policy gradient methods define a score function $J(\pi_\theta)$ of some parameterized (and differentiable) policy $\pi_\theta$ with parameters $\theta$, and use gradient ascent directly on $J(\pi_\theta)$ to update $\theta$.

There have been several recent successful applications of policy gradient algorithms in complex domains such as self-play learning in AlphaGo [80], Atari and 3D maze navigation [59], continuous control problems [76, 54, 21], robotics [27], and autonomous driving [78]. At the core of several recent state-of-the-art Deep RL algorithms [37, 22] is the advantage actor-critic (A2C) algorithm defined in [59]. In addition to learning a policy (*actor*), A2C learns a parameterized *critic*: an estimate of $v_\pi(s)$, which it then uses both to estimate the remaining return after $k$ steps, and as a control variate (*i.e.* baseline) that reduces the variance of the return estimates.

### 2.2  Game Theory, Regret Minimization, and Multiagent Reinforcement Learning

In multiagent RL (MARL), $n = |\mathcal{N}| = |\{1, 2, \cdots, n\}|$ agents interact within the same environment. At each step, each agent $i$ takes an action, and the joint action $\mathbf{a}$ leads to a new state $s_{t+1} \sim \mathcal{T}(s_t, \mathbf{a}_t)$; each player $i$ receives their own separate observation $o_i(s_t, \mathbf{a}, s_{t+1})$ and reward $r_{t,i}$. Each agent maximizes their own return $G_{t,i}$, or their expected return which depends on the joint policy $\pi$.

Much work in classical MARL focuses on Markov games where the environment is fully observable and agents take actions simultaneously, which in some cases admit Bellman operators [55, 103, 70, 69]. When the environment is partially observable, policies generally map to values and actions from agents' observation histories; even when the problem is cooperative, learning is hard [65].

We focus our attention to the setting of zero-sum games, where $\sum_{i \in \mathcal{N}} r_{t,i} = 0$. In this case, polynomial algorithms exist for finding optimal policies in finite tasks for the two-player case. The guarantees that Nash equilibrium provides are less clear for the $(n > 2)$-player case, and finding one is hard [20]. Despite this, regret minimization approaches are known to filter out dominated actions, and have empirically found good (*e.g.* competition-winning) strategies in this setting [74, 26, 48].

Partially observable environments require a few key definitions in order to define the notion of state. A **history** $h \in \mathcal{H}$ is a sequence of actions from all players *including the environment* taken from the start of an episode. The environment (also called "nature") is treated as a player with a fixed policy and there is a deterministic mapping from any $h$ to the actual state of the environment. Define an **information state**, $s_t = \{h \in \mathcal{H} \mid$ player $i$'s sequence of observations, $o_{i,t'<t}(s_{t'}, \mathbf{a}_{t'}, s_{t'+1})$, is consistent with $h\}$[4]. So, $s_t$ includes histories leading to $s_t$ that are indistinguishable to player $i$; *e.g.* in Poker, the $h \in s_t$ differ only in the private cards dealt to opponents. A joint policy $\pi$ is a **Nash equilibrium** if the incentive to deviate to a best response $\delta_i(\pi) = \max_{\pi_i'} \mathbb{E}_{\pi_i', \pi_{-i}}[G_{0,i}] - \mathbb{E}_\pi[G_{0,i}] = 0$ for each player $i \in \mathcal{N}$, where $\pi_{-i}$ is the set of $i's$ opponents' policies. Otherwise, $\epsilon$-equilibria are approximate, with $\epsilon = \max_i \delta_i(\pi)$. Regret minimization algorithms produce iterates whose average $\bar{\pi}$ reduces an upper bound on $\epsilon$, measured using $\text{NASHCONV}(\pi) = \sum_i \delta_i(\pi)$. Nash equilibrium is minimax-optimal in two-player zero-sum games: using one minimizes worst-case losses.

There are well-known links between learning, game theory and regret minimization [9]. One method, counterfactual regret (CFR) minimization [102], has led to significant progress in Poker AI. Let $\eta^\pi(h_t) = \prod_{t'<t} \pi(s_{t'}, a_{t'})$, where $h_{t'} \sqsubset h_t$ is a prefix, $h_{t'} \in s_{t'}, h_t \in s_t$, be the **reach probability** of $h$ under $\pi$ from all policies' action choices. This can be split into player $i$'s contribution and their opponents' (including nature's) contribution, $\eta^\pi(h) = \eta_i^\pi(h)\eta_{-i}^\pi(h)$. Suppose player $i$ is to play at $s$: under **perfect recall**, player $i$ remembers the sequence of their own states reached, which is the same for all $h \in s$, since they differ only in private information seen by opponent(s); as a result $\forall h, h' \in s, \eta_i^\pi(h) = \eta_i^\pi(h') := \eta_i^\pi(s)$. For some history $h$ and action $a$, we call $h$ a **prefix history** $h \sqsubset ha$, where $ha$ is the history $h$ followed by action $a$; they may also be smaller, so $h \sqsubset ha \sqsubset hab \Rightarrow h \sqsubset hab$. Let $\mathcal{Z} = \{z \in \mathcal{H} \mid z \text{ is terminal}\}$ and $\mathcal{Z}(s,a) = \{(h,z) \in \mathcal{H} \times \mathcal{Z} \mid h \in s, ha \sqsubseteq z\}$. CFR defines **counterfactual values** $v_i^c(\pi, s_t, a_t) = \sum_{(h,z) \in \mathcal{Z}(s_t, a_t)} \eta_{-i}^\pi(h)\eta_i^\pi(z)u_i(z)$, and $v_i^c(\pi, s_t) = \sum_a \pi(s_t, a)v_i^c(\pi, s_t, a_t)$, where $u_i(z)$ is the return to player $i$ along $z$, and accumulates regrets $\text{REG}_i(\pi, s_t, a') = v_i^c(\pi, s_t, a') - v_i^c(\pi, s_t)$, producing new policies from cumulative regret using *e.g.* regret-matching [28] or exponentially-weighted experts [6, 15].

CFR is a policy iteration algorithm that computes the expected values by visiting every possible trajectory, described in detail in Appendix B. Monte Carlo CFR (MCCFR) samples trajectories using an exploratory behavior policy, computing unbiased estimates $\hat{v}_i^c(\pi, s_t)$ and $\widehat{\text{REG}}_i(\pi, s_t)$ corrected by importance sampling [49]. Therefore, MCCFR is an *off-policy Monte Carlo* method. In one MCCFR variant, **model-free outcome sampling** (MFOS), the behavior policy at opponent states is defined as $\pi_{-i}$ enabling online regret minimization (player $i$ can update their policy independent of $\pi_{-i}$ and $\mathcal{T}$).

There are two main problems with (MC)CFR methods: (i) significant variance is introduced by sampling (off-policy) since quantities are divided by reach probabilities, (ii) there is no generalization across states except through expert abstractions and/or forward simulation with a perfect model. We show that actor-critics address both problems and that they are a form of *on-policy* MCCFR.

## 2.3 Most Closely Related Work

There is a rich history of policy gradient approaches in MARL. Early uses of gradient ascent showed that cyclical learning dynamics could arise, even in zero-sum matrix games [83]. This was partly addressed by methods that used variable learning rates [13, 11], policy prediction [99], and weighted updates [1]. The main limitation with these classical works was scalability: there was no direct way to use function approximation, and empirical analyses focused almost exclusively on one-shot games.

Recent work on policy gradient approaches to MARL addresses scalability by using newer algorithms such as A3C or TRPO [76]. However, they focus significantly less (if at all) on convergence guarantees. Naive approaches such as independent reinforcement learning fail to find optimal stochastic policies [55, 32] and can overfit the training data [50]. Much progress has been achieved for cooperative MARL: learning to communicate [51], Starcraft unit micromanagement [24], taxi fleet optimization [63], and autonomous driving [78]. There has also been progress for mixed cooperative/competitive environments: using a centralized critic [57], learning to negotiate [18], anticipating/learning opponent responses in social dilemmas [23, 53], and control in realistic physical environments [3, 7]. The most common methodology has been to train centrally (for decentralized execution), either having direct access to the other players' policy parameters or modeling them. As a result, assumptions are made about the other agents' policies, utilities, or learning mechanisms.

There are also methods that attempt to model the opponents [36, 30, 4]. Our methods do no such modeling, and can be classified in the "forget" category of the taxonomy proposed in [33]: that is, due to its on-policy nature, actors and critics adapt to and learn mainly from new/current experience.

We focus on the *model-free* (and online) setting: other agents' policies are inaccessible; training is not separated from execution. Actor-critics were recently studied in this setting for multiagent games [68], whereas we focus on partially-observable environments; only tabular methods are known to converge. Fictitious Self-Play computes approximate best responses via RL [31, 32], and can also be model-free. Regression CFR (RCFR) uses regression to estimate cumulative regrets from CFR [93]. RCFR is closely related to Advantage Regret Minimization (ARM) [38]. ARM [38] shows regret estimation methods handle partial observability better than standard RL, but was not evaluated in multiagent environments. In contrast, we focus primarily on the multiagent setting.

## 3   Multiagent Actor-Critics: Advantages and Regrets

CFR defines policy update rules from thresholded cumulative counterfactual regret: $\text{TCREG}_i(K, s, a) = (\sum_{k\in\{1,\cdots,K\}} \text{REG}_i(\pi_k, s, a))^+$, where $k$ is the number of iterations and $(x)^+ = \max(0, x)$. In CFR, regret matching updates a policy to be proportional to $\text{TCREG}_i(K, s, a)$.

On the other hand, REINFORCE [95] samples trajectories and computes gradients for each state $s_t$, updating $\boldsymbol{\theta}$ toward $\nabla_{\boldsymbol{\theta}} \log(s_t, a_t; \boldsymbol{\theta}) G_t$. A baseline is often subtracted from the return: $G_t - v_\pi(s_t)$, and policy gradients then become actor-critics, training $\pi$ and $v_\pi$ separately. The log appears due to the fact that action $a_t$ is sampled from the policy, the value is divided by $\pi(s_t, a_t)$ to ensure the estimate is properly estimating the true expectation [85, Section 13.3], and $\nabla_{\boldsymbol{\theta}} \pi(s_t, a_t; \boldsymbol{\theta})/\pi(s_t, a_t, \boldsymbol{\theta}) = \nabla_{\boldsymbol{\theta}} \log \pi(s_t, a_t; \boldsymbol{\theta})$. One could instead train $q_\pi$-based critics from states *and* actions. This leads to a $q$-based Policy Gradient (QPG) (also known as Mean Actor-Critic [5]):

$$\nabla_{\boldsymbol{\theta}}^{\text{QPG}}(s) = \sum_a [\nabla_\theta \pi(s, a; \boldsymbol{\theta})] \left( q(s, a; \mathbf{w}) - \sum_b \pi(s, b; \boldsymbol{\theta}) q(s, b, \mathbf{w}) \right), \qquad (1)$$

an advantage actor-critic algorithm differing from A2C in the (state-action) representation of the critics [56, 96] and summing over actions similar to the all-action algorithms [86, 71, 19, 5]. Interpreting $a_\pi(s, a) = q_\pi(s, a) - \sum_b \pi(s, b) q_\pi(s, b)$ as a regret, we can instead minimize a loss defined by an upper bound on the thresholded cumulative regret: $\sum_k (a_{\pi_k}(s, a))^+ \geq (\sum_k (a_{\pi_k}(s, a))^+$, moving the policy toward a no-regret region. We call this Regret Policy Gradient (RPG):

$$\nabla_{\boldsymbol{\theta}}^{\text{RPG}}(s) = -\sum_a \nabla_\theta \left( q(s, a; \mathbf{w}) - \sum_b \pi(s, b; \boldsymbol{\theta}) q(s, b; \mathbf{w}) \right)^+. \qquad (2)$$

The minus sign on the front represents a switch from gradient ascent on the score to *descent* on the loss. Another way to implement an adaptation of the regret-matching rule is by weighting the policy gradient by the thresholded regret, which we call Regret Matching Policy Gradient (RMPG):

$$\nabla_{\boldsymbol{\theta}}^{\text{RMPG}}(s) = \sum_a [\nabla_\theta \pi(s, a; \boldsymbol{\theta})] \left( q(s, a; \mathbf{w}) - \sum_b \pi(s, b; \boldsymbol{\theta}) q(s, b, \mathbf{w}) \right)^+. \qquad (3)$$

In each case, the critic $q(s_t, a_t; \mathbf{w})$ is trained in the standard way, using $\ell_2$ regression loss from sampled returns. The pseudo-code is given in Algorithm 2 in Appendix C. In Appendix F, we show that the QPG gradient is proportional to the RPG gradient at $s$: $\nabla_{\boldsymbol{\theta}}^{\text{RPG}}(s) \propto \nabla_{\boldsymbol{\theta}}^{\text{QPG}}(s)$.

### 3.1   Analysis of Learning Dynamics on Normal-Form Games

The first question is whether any of these variants can converge to an equilibrium, even in the simplest case. So, we now show phase portraits of the learning dynamics on Matching Pennies: a two-action version of Rock, Paper, Scissors. These analyses are common in multiagent learning as they allow visual depiction of the policy changes and how different factors affect the (convergence) behavior [83, 92, 13, 91, 11, 94, 1, 99, 98, 8, 89]. Convergence is difficult in Matching Pennies as the only Nash equilibrium $\pi^* = ((\frac{1}{2}, \frac{1}{2}), (\frac{1}{2}, \frac{1}{2}))$ requires learning stochastic policies. We give more detail and results on different games that cause cyclic learning behavior in Appendix D.

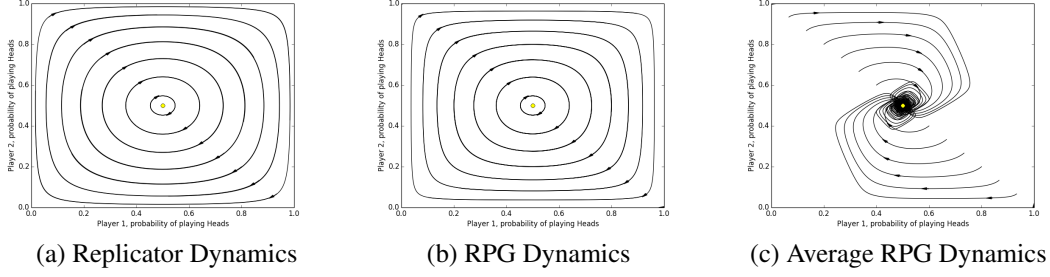

| (a) Replicator Dynamics | (b) RPG Dynamics | (c) Average RPG Dynamics |

Figure 1: Learning Dynamics in Matching Pennies: (a) and (b) show the vector field for $\partial \pi / \partial t$ including example particle traces, where each point is each player's probability of their first action; (c) shows example traces of policies following a discrete approximation to $\int_0^t \partial \pi / \partial t$.

In Figure 1, we see the similarity of the regret dynamics to replicator dynamics [88, 75]. We also show the *average policy dynamics* and observe convergence to equilibrium in each game we tried, which is a known to be guaranteed in two-player zero-sum games using CFR, fictitious play [14], and continuous replicator dynamics [35]. However, computing the average policy is complex [31, 102] and potentially worse with function approximation, requiring storing past data in large buffers [32].

## 3.2 Partially Observable Sequential Games

How do the values $v_i^c(\pi, s_t, a_t)$ and $q_{\pi,i}(s_t, a_t)$ differ? The authors of [38] posit that they are approximately equal when $s_t$ rarely occurs more than once in a trajectory. First, note that $s_t$ cannot be reached more than once in a trajectory from our definition of $s_t$, because the observation histories (of the player to play at $s_t$) would be different in each occurrence (*i.e.* due to perfect recall). So, the two values are indeed equal in deterministic, single-agent environments. In general, counterfactual values are conditioned on *player $i$ playing to reach $s_t$*, whereas $q$-function estimates are conditioned on *having reached $s_t$*. So, $q_{\pi,i}(s_t, a_t) = \mathbb{E}_{\rho \sim \pi}[G_{t,i} \mid S_t = s_t, A_t = a_t]$

$$= \sum_{h,z \in \mathcal{Z}(s_t, a_t)} \Pr(h \mid s_t) \eta^\pi(ha, z) u_i(z) \qquad \text{where } \eta^\pi(ha, z) = \frac{\eta^\pi(z)}{\eta^\pi(h)\pi(s,a)}$$

$$= \sum_{h,z \in \mathcal{Z}(s_t, a_t)} \frac{\Pr(s_t \mid h) \Pr(h)}{\Pr(s_t)} \eta^\pi(ha, z) u_i(z) \qquad \text{by Bayes' rule}$$

$$= \sum_{h,z \in \mathcal{Z}(s_t, a_t)} \frac{\Pr(h)}{\Pr(s_t)} \eta^\pi(ha, z) u_i(z) \qquad \text{since } h \in s_t, h \text{ is unique to } s_t$$

$$= \sum_{h,z \in \mathcal{Z}(s_t, a_t)} \frac{\eta^\pi(h)}{\sum_{h' \in s_t} \eta^\pi(h')} \eta^\pi(ha, z) u_i(z)$$

$$= \sum_{h,z \in \mathcal{Z}(s_t, a_t)} \frac{\eta_i^\pi(h) \eta_{-i}^\pi(h)}{\sum_{h' \in s_t} \eta_i^\pi(h') \eta_{-i}^\pi(h')} \eta^\pi(ha, z) u_i(z)$$

$$= \sum_{h,z \in \mathcal{Z}(s_t, a_t)} \frac{\eta_i^\pi(s) \eta_{-i}^\pi(h)}{\eta_i^\pi(s) \sum_{h' \in s_t} \eta_{-i}^\pi(h')} \eta^\pi(ha, z) u_i(z) \quad \text{due to def. of } s_t \text{ and perfect recall}$$

$$= \sum_{h,z \in \mathcal{Z}(s_t, a_t)} \frac{\eta_{-i}^\pi(h)}{\sum_{h' \in s_t} \eta_{-i}^\pi(h')} \eta^\pi(ha, z) u_i(z) = \frac{1}{\sum_{h \in s_t} \eta_{-i}^\pi(h)} v_i^c(\pi, s_t, a_t).$$

The derivation is similar to show that $v_{\pi,i}(s_t) = v_i^c(\pi, s_t) / \sum_{h \in s_t} \eta_{-i}^\pi(h)$. Hence, counterfactual values and standard value functions are generally not equal, but are scaled by the Bayes normalizing constant $\mathcal{B}_{-i}(\pi, s_t) = \sum_{h \in s_t} \eta_{-i}^\pi(h)$. If there is a low probability of reaching $s_t$ due to the environment or due to opponents' policies, these values will differ significantly.

This leads to a new interpretation of actor-critic algorithms in the multiagent partially observable setting: the advantage values $q_{\pi,i}(s_t, a_t) - v_{\pi,i}(s_t, a_t)$ are immediate counterfactual regrets scaled by $1/\mathcal{B}_{-i}(\pi, s_t)$. This then determines requirements for convergence guarantees in the tabular case.

Note that the standard policy gradient theorem holds: gradients can be estimated from samples. This follows from the derivation of the policy gradient in the tabular case (see Appendix E). When TD bootstrapping is not used, the Markov property is not required; having multiple agents and/or partial observability does not change this. For a proof using REINFORCE ($G_t$ only), see [78, Theorem 1]. The proof trivially follows using $G_{t,i} - v_{\pi,i}$ since $v_{\pi,i}$ is trained separately and does not depend on $\rho$.

Policy gradient algorithms perform gradient ascent on $J^{PG}(\pi_{\boldsymbol{\theta}}) = v_{\pi_\theta}(s_0)$, using $\nabla_{\boldsymbol{\theta}} J^{PG}(\pi_{\boldsymbol{\theta}}) \propto \sum_s \mu(s) \sum_a \nabla_{\boldsymbol{\theta}} \pi_\theta(s, a) q_\pi(s, a)$, where $\mu$ is on-policy distribution under $\pi$ [85, Section 13.2]. The actor-critic equivalent is $\nabla_{\boldsymbol{\theta}} J^{AC}(\pi_{\boldsymbol{\theta}}) \propto \sum_s \mu(s) \sum_a \nabla_{\boldsymbol{\theta}} \pi_\theta(s, a)(q_\pi(s, a) - \sum_b \pi(s, b) q_\pi(s, b))$. Note that the baseline is unnecessary when summing over the actions and $\nabla_{\boldsymbol{\theta}} J^{AC}(\pi_{\boldsymbol{\theta}}) = \nabla_{\boldsymbol{\theta}} J^{PG}(\pi_{\boldsymbol{\theta}})$ [5]. However, our analysis relies on a projected gradient descent algorithm that does not assume simplex constraints on the policy: in that case, in general $\nabla_{\boldsymbol{\theta}} J^{AC}(\pi_{\boldsymbol{\theta}}) \neq \nabla_{\boldsymbol{\theta}} J^{PG}(\pi_{\boldsymbol{\theta}})$.

**Definition 1.** *Define **policy gradient policy iteration** (PGPI) as a process that iteratively runs $\boldsymbol{\theta} \leftarrow \boldsymbol{\theta} + \alpha \nabla_{\boldsymbol{\theta}} J^{PG}(\pi_{\boldsymbol{\theta}})$, and **actor-critic policy iteration** (ACPI) similarly using $\nabla_{\boldsymbol{\theta}} J^{AC}(\pi_{\boldsymbol{\theta}})$.*

In two-player zero-sum games, PGPI/ACPI are gradient ascent-descent problems, because each player is trying to ascend their own score function, and when using tabular policies a solution exists due to the minimax theorem [79]. Define player $i$'s **external regret** over $K$ steps as $R_i^K = \max_{\pi_i' \in \Pi_i} \left( \sum_{k=1}^{K} \mathbb{E}_{\pi_i'}[G_{0,i}] - \mathbb{E}_{\pi^k}[G_{0,i}] \right)$, where $\Pi_i$ is the set of deterministic policies.

**Theorem 1.** *In two-player zero-sum games, when using tabular policies and an $\ell_2$ projection $P(\boldsymbol{\theta}) = \operatorname{argmin}_{\boldsymbol{\theta}' \in \Delta(\mathcal{S}, \mathcal{A})} \|\boldsymbol{\theta} - \boldsymbol{\theta}'\|_2$, where $\Delta(\mathcal{S}, \mathcal{A}) = \{\boldsymbol{\theta} \mid \forall s \in \mathcal{S}, \sum_{b \in \mathcal{A}} \boldsymbol{\theta}_{s,b} = 1\}$ is the space of tabular simplices, if player $i$ uses learning rates of $\alpha_{s,k} = k^{-\frac{1}{2}} \eta_i^{\pi^k}(s) \mathcal{B}_{-i}(\pi, s_t)$ at $s$ on iteration $k$, and $\theta_{s,a}^k > 0$ for all $k$ and $s$, then projected PGPI, $\theta_{s,\cdot}^{k+1} \leftarrow P(\{\theta_{s,a}^k + \alpha_{s,k} \frac{\partial}{\partial \theta_{s,a}^k} J^{PG}(\pi_{\boldsymbol{\theta}^k})\}_a)$, has regret $R_i^K \leq \frac{1}{\eta_i^{\min}} |\mathcal{S}_i| \left( \sqrt{K} + (\sqrt{K} - \frac{1}{2}) |\mathcal{A}| (\Delta r)^2 \right)$, where $\mathcal{S}_i$ is the set of player $i$'s states, $\Delta r$ is the reward range, and $\eta_i^{\min} = \min_{s,k} \eta_i^k(s)$. The same holds for projected ACPI (see appendix).*

The proof is given in Appendix E. In the case of sampled trajectories, as long as every state is reached with positive probability, Monte Carlo estimators of $q_{\pi,i}$ will be consistent. Therefore, we use exploratory policies and decay exploration over time. With a finite number of samples, the probability that an estimator $\hat{q}_{\pi,i}(s, a)$ differs by some quantity away from its mean is determined by Hoeffding's inequality and the reach probabilities. We suspect these errors could be accumulated to derive probabilistic regret bounds similar to the off-policy Monte Carlo case [46].

What happens in the sampling case with a fixed per-state learning rate $\alpha_s$? If player $i$ collects a batch of data from many sampled episodes and applies them all at once, then the *effective* learning rates (expected update rate relative to the other states) is scaled by the probability of reaching $s$: $\eta_i^\pi(s) \mathcal{B}_{-i}(\pi, s)$, which matches the value in the condition of Theorem 1. This suggests using a globally decaying learning rate to simulate the remaining $k^{-\frac{1}{2}}$.

The analysis so far has concentrated on establishing guarantees for the optimization problem that underlies standard formulation of policy gradient and actor-critic algorithms. A better guarantee can be achieved by using stronger policy improvement (proof and details are found in Appendix E):

**Theorem 2.** *Define a state-local $J^{PG}(\pi_{\boldsymbol{\theta}}, s) = v_{\pi_{\boldsymbol{\theta}}, i}(s)$, composite gradient $\{\frac{\partial}{\partial \theta_{s,a}} J^{PG}(\pi_{\boldsymbol{\theta}}, s)\}_{s,a}$, **strong policy gradient policy iteration** (SPGPI), and **strong actor-critic policy iteration** (SACPI) as in Definition 1 except replacing the gradient components with $\frac{\partial}{\partial \theta_{s,a}} J^{PG}(\pi_{\boldsymbol{\theta}}, s)$. Then, in two-player zero-sum games, when using tabular policies and projection $P(\boldsymbol{\theta})$ as defined in Theorem 1 with learning rates $\alpha_k = k^{-\frac{1}{2}}$ on iteration $k$, projected SPGPI, $\theta_{s,\cdot}^{k+1} \leftarrow P(\{\theta_{s,a}^k + \alpha_k \frac{\partial}{\partial \theta_{s,a}^k} J^{PG}(\pi_{\boldsymbol{\theta}}, s)\}_a)$, has regret $R_i^K \leq |\mathcal{S}_i| \left( \sqrt{K} + (\sqrt{K} - \frac{1}{2}) |\mathcal{A}| (\Delta r)^2 \right)$, where $\mathcal{S}_i$ is the set of player $i$'s states and $\Delta r$ is the reward range. This also holds for projected SACPI (see appendix).*

# 4 Empirical Evaluation

We now assess the behavior of the actor-critic algorithms in practice. While the analyses in the previous section established guarantees for the tabular case, ultimately we want to assess scalability and generalization potential for larger settings. Our implementation parameterizes critics and policies using neural networks with two fully-connected layers of 128 units each, and rectified linear unit activation functions, followed by a linear layer to output a single value $q$ or softmax layer to output $\pi$. We chose these architectures to remain consistent with previous evaluations [32, 50].

## 4.1 Domains: Kuhn and Leduc Poker

We evaluate the actor-critic algorithms on two $n$-player games: Kuhn poker, and Leduc poker.

**Kuhn poker** is a toy game where each player starts with 2 chips, antes 1 chip to play, and receives one card face down from a deck of size $n + 1$ (one card remains hidden). Players proceed by betting (raise/call) by adding their remaining chip to the pot, or passing (check/fold) until all players are either in (contributed as all other players to the pot) or out (folded, passed after a raise). The player with the highest-ranked card that has not folded wins the pot.

In **Leduc poker**, players have a limitless number of chips, and the deck has size $2(n + 1)$, divided into two suits of identically-ranked cards. There are two rounds of betting, and after the first round a single public card is revealed from the deck. Each player antes 1 chip to play, and the bets are limited to two per round, and number of chips limited to 2 in the first round, and 4 in the second round.

The rewards to each player is the number of chips they had after the game minus before the game. To remain consistent with other baselines, we use the form of Leduc described in [50] which does not restrict the action space, adding reward penalties if/when illegal moves are chosen.

## 4.2 Baseline: Neural Fictitious Self-Play

We compare to one main baseline. **Neural Fictitious Self-Play** (NFSP) is an implementation of fictitious play, where approximate best responses are used in place of full best response [32]. Two transition buffers of are used: $\mathcal{D}^{RL}$ and $\mathcal{D}^{ML}$; the former to train a DQN agent towards a best response $\pi_i$ to $\bar{\pi}_{-i}$, data in the latter is replaced using reservoir sampling, and trains $\bar{\pi}_i$ by classification.

## 4.3 Main Performance Results

Here we show the empirical convergence to approximate Nash equlibria for each algorithm in self-play, and performance against fixed bots. The standard metric to use for this is NASHCONV($\pi$) defined in Section 2.2, which reports the accuracy of the approximation to a Nash equilibrium.

**Training Setup**. In the domains we tested, we observed that the variance in returns was high and hence we performed multiple policy evaluation updates ($q$-update for $\nabla^{\text{QPG}}$, $\nabla^{\text{RPG}}$, and $\nabla^{\text{RMPG}}$, and $v$-update for A2C) followed by policy improvement (policy gradient update). These updates were done using separate SGD optimizers with their respective learning rates of fixed 0.001 for policy evaluation, and annealed from a starting learning rate to 0 over 20M steps for policy improvement. (See Appendix G for exact values). Further, the policy improvement step is applied after $N_q$ policy evaluation updates. We treat $N_q$ and batch size as a hyper parameters and sweep over a few reasonable values. In order to handle different scales of rewards in the multiple domains, we used the streaming Z-normalization on the rewards, inspired by its use in Proximal Policy Optimization (PPO) [77]. In addition, the agent's policy is controlled by a(n inverse) temperature added as part of the softmax operator. The temperature is annealed from 1 to 0 over 1M steps to ensure adequate state space coverage. An additional entropy cost hyper-parameter is added as is standard practice with Deep RL policy gradient methods such as A3C [59, 77]. For NFSP, we used the same values presented in [50].

**Convergence to Equilibrium.** See Figure 2 for convergence results. Please note that we plot the NASHCONV for the average policy in the case of NFSP, and the current policy in the case of the policy gradient algorithms. We see that in 2-player Leduc, the actor-critic variants we tried are similar in performance; NFSP has faster short-term convergence but long-term the actor critics are comparable. Each converges significantly faster than A2C. However RMPG seems to plateau.

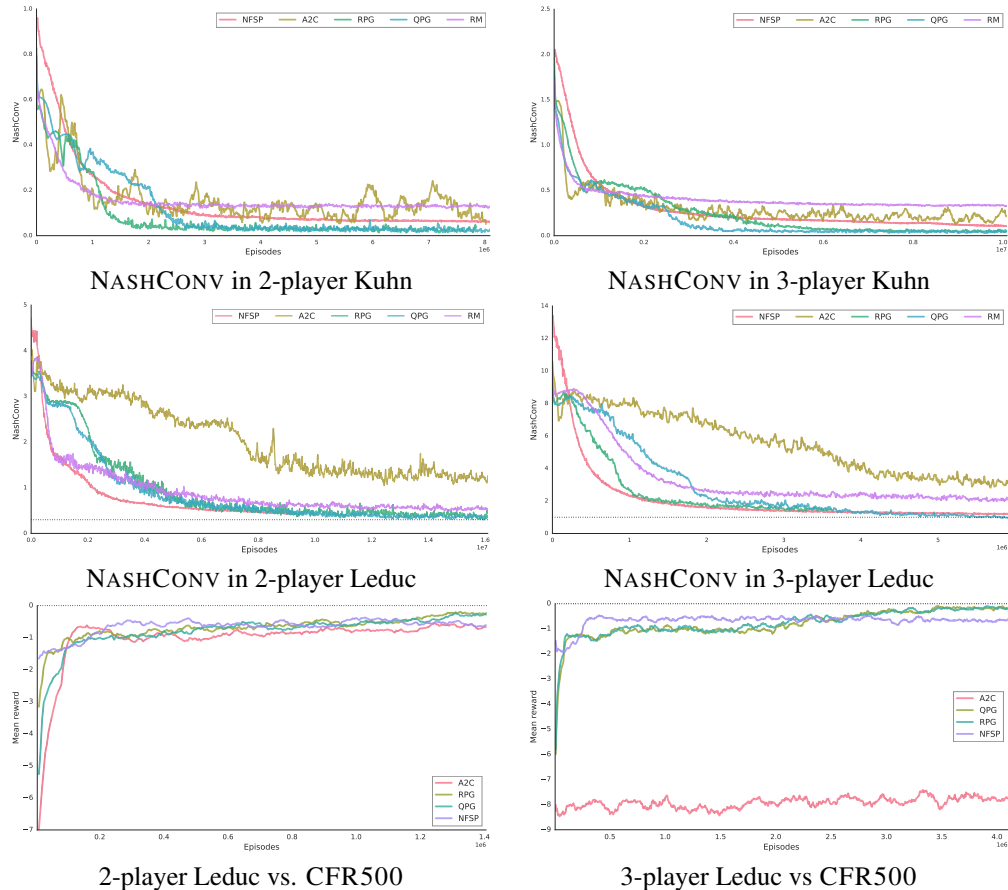

NASHCONV in 2-player Kuhn

NASHCONV in 3-player Kuhn

NASHCONV in 2-player Leduc

NASHCONV in 3-player Leduc

2-player Leduc vs. CFR500

3-player Leduc vs CFR500

Figure 2: Empirical convergence rates for NASHCONV($\pi$) and performance versus CFR agents.

**Performance Against Fixed Bots.** We also measure the expected reward against fixed bots, averaged over player seats. These bots, CFR500, correspond to the average policy after 500 iterations of CFR. QPG and RPG do well here, scoring higher than A2C and even beating NFSP in the long-term.

# 5 Conclusion

In this paper, we discuss several update rules for actor-critic algorithms in multiagent reinforcement learning. One key property of this class of algorithms is that they are model-free, leading to a purely online algorithm, independent of the opponents and environment. We show a connection between these algorithms and (counterfactual) regret minimization, leading to previously unknown convergence properties underlying model-free MARL in zero-sum games with imperfect information.

Our experiments show that these actor-critic algorithms converge to approximate Nash equilibria in commonly-used benchmark Poker domains with rates similar to or better than baseline model-free algorithms for zero-sum games. However, they may be easier to implement, and do not require storing a large memory of transitions. Furthermore, the current policy of some variants do significantly better than the baselines (including the average policy of NFSP) when evaluated against fixed bots. Of the actor-critic variants, RPG and QPG seem to outperform RMPG in our experiments.

As future work, we would like to formally develop the (probabilistic) guarantees of the sample-based on-policy Monte Carlo CFR algorithms and/or extend to continuing tasks as in MDPs [41]. We are also curious about what role the connections between actor-critic methods and CFR could play in deriving convergence guarantees in model-free MARL for cooperative and/or potential games.

**Acknowledgments.** We would like to thank Martin Schmid, Audrūnas Gruslys, Neil Burch, Noam Brown, Kevin Waugh, Rich Sutton, and Thore Graepel for their helpful feedback and support.

## Footnotes

[1]Appendices are included in the technical report version of the paper; see [84].

[2]Note that in fully-observable settings, $o(s_t, a_t, s_{t+1}) = s_{t+1}$. In partially observable environments [39, 65], an observation function $\mathcal{O} : \mathcal{S} \times \mathcal{A} \rightarrow \Delta(\Omega)$ is used to sample $o(s_t, a_t, s_{t+1}) \sim O(s_t, a_t)$.

[3] We assume finite episodic tasks of bounded length and leave out the discount factor $\gamma$ to simplify the notation, without loss of generality. We use $\gamma(= 0.99)$-discounted returns in our experiments.

[4]In defining $s_t$, we drop the reference to acting player $i$ in turn-based games without loss of generality.

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
