[Reviews · NeurIPS 2018]

Reviewer 1



2. The paper draws the connection between regret minimization and actor-critic algorithms. Specifically, it shows the connection by defining a new variant of an actor-critic algorithm that performs an exhaustive policy evaluation at each stage (denoted as policy-iteration-actor-critic), together with an adaptive learning rate. Then, under this setting, it is said that the actor-critic algorithm basically minimizes regret and converges to a Nash equilibrium. The paper suggests a few new versions of policy gradient update rules (Q-based Policy Gradient, Regret Policy Gradient, and Regret Matching Policy Gradient) and evaluates them on multi-agent zero-sum imperfect information games. To my understanding, Q-Based Policy Gradient is basically an advantage actor-critic algorithm (up to a transformation of the learned baseline) 3. The authors mention a “reasonable parameter sweep” over the hyperparameters. I’m curious to know the stability of the proposed actor-critic algorithms over the different trials 4. The paper should be proofread again. Many broken sentences and typos (“…be the reach probability of the all policies action choices…”, “…Two transition buffers of are used…”) 5. G_t is not defined. I believe the meaning is cumulative reward R_t, since the authors use them interchangeably. 6. Being at the border between game theory and RL makes notation very confusing. For example, pi that stands for policy (a.k.a strategy in game theory literature) denotes “transition probability” in game theory. Therefore, I encourage the authors to include a table of notations. 7. In line 148 the authors give a fresh intuition about the policy gradient expression (“…so estimate of the gradient must be corrected by dividing by policy probability”). However, that’s counterintuitive. A sampled trajectory with low probabilities would produce a high-intensity gradient (dividing by small numbers), which is the opposite of what is meant to happen. Please clarify your intuition. 8. In line 150 you define Q-based policy gradient, which basically has the form of advantage actor-critic. If this is correct, then please illuminate the relation for the sake of clarity (little or none knows what Q-based PG is vs the popularity of advantage actor-critic methods) 9. Please explain the untraditional containment symbol of ha and z in line 187. Did you mean to indicate set containment? And please clarify what is “ha” (you’ve defined “eta of ha and z” but never “ha” explicitly) 10. Do you share the same critic for all players? Summary: Quality - the paper draws a connection between Q values in RL and counterfactual values from the field of game theory. Second, it draws a connection between actor-critic and regret minimization. Although this is true under a very specific setup, I believe that this is an important result that connects two worlds that have been parallel up to now. Clarity - The paper is hard to follow because it stands on the borderline between disciplines with conflicting notations. In order to remedy the problem, the paper should pay extra attention to notations and their use. Besides that, an extra proofreading of the paper would benefit the paper significantly (many broken sentences and typos) Originality and significance - I believe that the paper is novel and significant to a fair degree

Reviewer 2



Summary: The paper presents a connection between actor-critic algorithms (usually framed in single-agent RL) and counterfactual regret minimization (framed for multiagent learning). The results show that the counterfactual values and the standard value functions are related (scaled by a normalization constant) and that advantage values are immediate counterfactual regrets (scaled by another constant). Experiments in two poker versions show that the algorithms converge close to the Nash equulibrium in self-play and also works well against other opponents (bots). Comments: I appreciate the thorough revision of related work, if anything I must comment on a few recent works that are relevant. Two recent surveys on agents modeling agents and dealing with non-stationarity in multiagent interactions. -S. V. Albrecht and P. Stone, “Autonomous agents modelling other agents: A comprehensive survey and open problems” -P. Hernandez-Leal, M. Kaisers, T. Baarslag, and E. Munoz de Cote, “A Survey of Learning in Multiagent Environments - Dealing with Non-Stationarity” A recent work on Deep MARL. -Z. Hong, S. Su, T. Shann, Y. Chang, and C. Lee "A Deep Policy Inference Q-Network for Multi-Agent Systems" I would like if you can comment more on the implications of 3.2, in particular about "If there is a low probability of reaching s_t due to the environment or due to opponents' policies, these values will differ significantly" It is not clear why two transition buffers are needed (in 4.2) Post rebuttal: The work shows a nice connection between two areas. I strongly recommend the authors to improve the presentation: - background (CFR and recent papers), - contrasting with well-known algorithms (A2C), - check carefully notation - solve typos and broken sentences.

Reviewer 3



In this paper the authors consider actor-critic algorithms for POMDPs in a multiagent setting. They derive policy update rules, prove convergence properties of those and evaluate those learned policies empirically. I am not an expert in the field of multi-agent RL but got the feeling that the presented results are novel and extend current knowledge in an interesting and valuable way. Although there is no clear advantage of the proposed algorithms empirically, I think the authors make a sensible step forward by enabling the use of model-free algorithms for this multi-agent setting under partial observability. The paper could be improved by cleaning up notation (e.g. u_i in line 87, REG without subscripts, symbol in line 187 .... are never defined --- their definitions following implicitly). I would also recommend to give some more details on CFR to make the paper more self contained as I had a hard time following without referring back to the original paper. Furthermore it would make the paper more compelling if the proposed algorithms were studied in more detail, e.g. highlighting sensitivity to the hyperparameters. For instance, in section 4.4 the authors comment on the need to interleave policy improvements with multiple updates to the critic. It would be helpful to report the authors precise findings (and the actually used update frequencies). -- I have read the author response and the authors have clarified my questions. Upon another reading of the paper and the other reviews, I decided to increase my score to reflect the relevance of the paper.